# Extremal Mechanisms for Local Differential Privacy

**Peter Kairouz**[1]  **Sewoong Oh**[2]  **Pramod Viswanath**[1]
[1]Department of Electrical & Computer Engineering
[2]Department of Industrial & Enterprise Systems Engineering
University of Illinois Urbana-Champaign
Urbana, IL 61801, USA
{kairouz2,swoh,pramodv}@illinois.edu

## Abstract

Local differential privacy has recently surfaced as a strong measure of privacy in contexts where personal information remains private even from data analysts. Working in a setting where the data providers and data analysts want to maximize the utility of statistical inferences performed on the released data, we study the fundamental tradeoff between local differential privacy and information theoretic utility functions. We introduce a family of extremal privatization mechanisms, which we call staircase mechanisms, and prove that it contains the optimal privatization mechanism that maximizes utility. We further show that for all information theoretic utility functions studied in this paper, maximizing utility is equivalent to solving a linear program, the outcome of which is the optimal staircase mechanism. However, solving this linear program can be computationally expensive since it has a number of variables that is exponential in the data size. To account for this, we show that two simple staircase mechanisms, the binary and randomized response mechanisms, are universally optimal in the high and low privacy regimes, respectively, and well approximate the intermediate regime.

## 1 Introduction

In statistical analyses involving data from individuals, there is an increasing tension between the need to share the data and the need to protect sensitive information about the individuals. For example, users of social networking sites are increasingly cautious about their privacy, but still find it inevitable to agree to share their personal information in order to benefit from customized services such as recommendations and personalized search [1, 2]. There is a certain utility in sharing data for both data providers and data analysts, but at the same time, individuals want *plausible deniability* when it comes to sensitive information.

For such systems, there is a natural core optimization problem to be solved. Assuming both the data providers and analysts want to maximize the utility of the released data, how can they do so while preserving the privacy of participating individuals? The formulation and study of an optimal framework addressing this tradeoff is the focus of this paper.

**Local differential privacy.** The need for data privacy appears in two different contexts: the *local privacy* context, as in when individuals disclose their personal information (e.g., voluntarily on social network sites), and the *global privacy* context, as in when institutions release databases of information of several people or answer queries on such databases (e.g., US Government releases census data, companies like Netflix release proprietary data for others to test state of the art data analytics). In both contexts, privacy is achieved by *randomizing* the data before releasing it. We study the setting of local privacy, in which data providers do not trust the data collector (analyst). Local privacy dates back to Warner [29], who proposed the *randomized response* method to provide plausible deniability for individuals responding to sensitive surveys.

A natural notion of privacy protection is making inference of information beyond what is released hard. *Differential privacy* has been proposed in the global privacy context to formally capture this notion of privacy [11, 13, 12]. In a nutshell, differential privacy ensures that an adversary should not be able to reliably infer whether or not a particular individual is participating in the database query, even with unbounded computational power and access to every entry in the database except for that particular individual's data. Recently, the notion of differential privacy has been extended to the local privacy context [10]. Formally, consider a setting where there are $n$ data providers each owning a data $X_i$ defined on an input alphabet $\mathcal{X}$. In this paper, we shall deal, almost exclusively, with finite alphabets. The $X_i$'s are independently sampled from some distribution $P_\nu$ parameterized by $\nu \in \{0, 1\}$. A statistical privatization mechanism $Q_i$ is a conditional distribution that maps $X_i \in \mathcal{X}$ stochastically to $Y_i \in \mathcal{Y}$, where $\mathcal{Y}$ is an output alphabet possibly larger than $\mathcal{X}$. The $Y_i$'s are referred to as the privatized (sanitized) views of $X_i$'s. In a non-interactive setting where the individuals do not communicate with each other and the $X_i$'s are independent and identically distributed, the same privatization mechanism $Q$ is used by all individuals. For a non-negative $\varepsilon$, we follow the definition of [10] and say that a mechanism $Q$ is $\varepsilon$-*locally differentially private* if

$$\sup_{S \in \sigma(\mathcal{Y}), x, x' \in \mathcal{X}} \frac{Q(S|X_i = x)}{Q(S|X_i = x')} \leq e^\varepsilon , \tag{1}$$

where $\sigma(\mathcal{Y})$ denotes an appropriate $\sigma$-field on $\mathcal{Y}$.

**Information theoretic utilities for statistical analyses.** The data analyst is interested in the *statistics* of the data as opposed to individual samples. Naturally, the utility should also be measured in terms of the distribution rather than sample quantities. Concretely, consider a client-server setting, where each client with data $X_i$ sends a privatized version of the data $Y_i$, via an $\varepsilon$-locally differentially private privatization mechanism $Q$. Given the privatized views $\{Y_i\}_{i=1}^n$, the data analyst wants to make inferences based on the induced marginal distribution

$$M_\nu(S) \equiv \int Q(S|x)dP_\nu(x) , \tag{2}$$

for $S \in \sigma(\mathcal{Y})$ and $\nu \in \{0, 1\}$. The power to discriminate data generated from $P_0$ to data generated from $P_1$ depends on the 'distance' between the marginals $M_0$ and $M_1$. To measure the ability of such statistical discrimination, our choice of utility of a particular privatization mechanism $Q$ is an information theoretic quantity called Csiszár's $f$-divergence defined as

$$D_f(M_0||M_1) = \int f\left(\frac{dM_0}{dM_1}\right) dM_1 , \tag{3}$$

for some convex function $f$ such that $f(1) = 0$. The Kullback-Leibler (KL) divergence $D_{\mathrm{kl}}(M_0||M_1)$ is a special case with $f(x) = x \log x$, and so is the total variation $\|M_0 - M_1\|_{\mathrm{TV}}$ with $f(x) = (1/2)|x - 1|$. Such $f$-divergences capture the quality of statistical inference, such as minimax rates of statistical estimation or error exponents in hypothesis testing [28]. As a motivating example, suppose a data analyst wants to test whether the data is generated from $P_0$ or $P_1$ based on privatized views $Y_1, \ldots, Y_n$. According to Chernoff-Stein's lemma, for a bounded type I error probability, the best type II error probability scales as $e^{-n \, D_{\mathrm{kl}}(M_0||M_1)}$. Naturally, we are interested in finding a privatization mechanism $Q$ that minimizes the probability of error by solving the following constraint maximization problem

$$\underset{Q \in \mathcal{D}_\varepsilon}{\text{maximize}} \qquad D_{\mathrm{kl}}(M_0||M_1) , \tag{4}$$

where $\mathcal{D}_\varepsilon$ is the set of all $\varepsilon$-locally differentially private mechanisms satisfying (1). Motivated by such applications in statistical inference, our goal is to provide a general framework for finding optimal privatization mechanisms that maximize the $f$-divergence between the induced marginals under local differential privacy.

**Contributions.** We study the fundamental tradeoff between local differential privacy and $f$-divergence utility functions. The privacy-utility tradeoff is posed as a constrained maximization problem: maximize $f$-divergence utility functions subject to local differential privacy constraints. This maximization problem is (a) nonlinear: $f$-divergences are convex in $Q$; (b) non-standard: we are maximizing instead of minimizing a convex function; and (c) infinite dimensional: the space of all differentially private mechanisms is uncountable. We show, in Theorem 2.1, that for all $f$-divergences, any $\varepsilon$, and any pair of distributions $P_0$ and $P_1$, a *finite* family of *extremal* mechanisms

(a subset of the corner points of the space of privatization mechanisms), which we call *staircase* mechanisms, contains the optimal privatization mechanism. We further prove, in Theorem 2.2, that solving the original problem is equivalent to solving a linear program, the outcome of which is the optimal staircase mechanism. However, solving this linear program can be computationally expensive since it has $2^{|\mathcal{X}|}$ variables. To account for this, we show that two simple staircase mechanisms (the binary and randomized response mechanisms) are optimal in the high and low privacy regimes, respectively, and well approximate the intermediate regime. This contributes an important progress in the differential privacy area, where the privatization mechanisms have been few and almost no exact optimality results are known. As an application, we show that the effective sample size reduces from $n$ to $\varepsilon^2 n$ under local differential privacy in the context of hypothesis testing.

**Related work.** Our work is closely related to the recent work of [10] where an upper bound on $D_{\mathrm{kl}}(M_0 \| M_1)$ was derived under the same local differential privacy setting. Precisely, Duchi et. al. proved that the KL-divergence maximization problem in (4) is at most $4(e^\varepsilon - 1)^2 \|P_1 - P_2\|_{TV}^2$. This bound was further used to provide a minimax bound on statistical estimation using information theoretic converse techniques such as Fano's and Le Cam's inequalities.

In a similar spirit, we are also interested in maximizing information theoretic quantities of the marginals under local differential privacy. We generalize the results of [10], and provide stronger results in the sense that we $(a)$ consider a broader class of information theoretic utilities; $(b)$ provide explicit constructions of the optimal mechanisms; and $(c)$ recover the existing result of [10, Theorem 1] (with a stronger condition on $\varepsilon$).

While there is a vast literature on differential privacy, exact optimality results are only known for a few cases. The typical recipe is to propose a differentially private mechanism inspired by [11, 13, 26, 20], and then establish its near-optimality by comparing the achievable utility to a converse, for example in principal component analysis [8, 5, 19, 24], linear queries [21, 18], logistic regression [7] and histogram release [25]. In this paper, we take a different route and solve the utility maximization problem *exactly*.

Optimal differentially private mechanisms are known only in a few cases. Ghosh et. al. showed that the geometric noise adding mechanism is optimal (under a Bayesian setting) for monotone utility functions under count queries (sensitivity one) [17]. This was generalized by Geng et. al. (for a worst-case input setting) who proposed a family of mechanisms and proved its optimality for monotone utility functions under queries with arbitrary sensitivity [14, 16, 15]. The family of optimal mechanisms was called *staircase mechanisms* because for any $y$ and any neighboring $x$ and $x'$, the ratio of $Q(y|x)$ to $Q(y|x')$ takes one of three possible values $e^\varepsilon$, $e^{-\varepsilon}$, or 1. Since the optimal mechanisms we develop also have an identical property, we retain the same nomenclature.

## 2  Main results

In this section, we give a formal definition for staircase mechanisms and show that they are the optimal solutions to maximization problems of the form (5). Using the structure of staircase mechanisms, we propose a combinatorial representation for this family of mechanisms. This allows us to reduce the nonlinear program of (5) to a linear program with $2^{|\mathcal{X}|}$ variables. Potentially, for any instance of the problem, one can solve this linear program to obtain the optimal privatization mechanism, albeit with significant computational challenges since the number of variables scales exponentially in the alphabet size. To address this, we prove that two simple staircase mechanisms, which we call the binary mechanism and the randomized response mechanism, are optimal in high and low privacy regimes, respectively. We also show how our results can be used to derive upper bounds on $f$-divergences under privacy. Finally, we give a concrete example illustrating the exact tradeoff between privacy and statistical inference in the context of hypothesis testing.

### 2.1  Optimality of staircase mechanisms

Consider a random variable $X \in \mathcal{X}$ generated according to $P_\nu$, $\nu \in \{0, 1\}$. The distribution of the privatized output $Y$, whenever $X$ is distributed according to $P_\nu$, is represented by $M_\nu$ and given by (2). We are interested in characterizing the optimal solution of

$$\underset{Q \in \mathcal{D}_\varepsilon}{\text{maximize}} \qquad D_f(M_0 \| M_1) \,, \tag{5}$$

where $\mathcal{D}_\varepsilon$ is the set of all $\varepsilon$-differentially private mechanisms satisfying, for all $x, x' \in \mathcal{X}$ and $y \in \mathcal{Y}$,

$$0 \leq \left| \ln \left( \frac{Q(y|x)}{Q(y|x')} \right) \right| \leq \varepsilon . \tag{6}$$

This includes maximization over information theoretic quantities of interest in statistical estimation and hypothesis testing such as total variation, KL-divergence, and $\chi^2$-divergence [28]. In general this is a complicated nonlinear program: we are maximizing a convex function in $Q$; further, the dimension of $Q$ might be unbounded: the optimal privatization mechanism $Q^*$ might produce an infinite output alphabet $\mathcal{Y}$. The following theorem proves that one never needs an output alphabet larger than the input alphabet in order to achieve the maximum divergence, and provides a combinatorial representation of the optimal solution.

**Theorem 2.1.** *For any $\varepsilon$, any pair of distributions $P_0$ and $P_1$, and any $f$-divergence, there exists an optimal mechanism $Q^*$ maximizing the $f$-divergence in* (5) *over all $\varepsilon$-locally differentially private mechanisms, such that*

$$\left| \ln \left( \frac{Q^*(y|x)}{Q^*(y|x')} \right) \right| \in \{0, \varepsilon\} , \tag{7}$$

*for all $y \in \mathcal{Y}, x, x' \in \mathcal{X}$ and the output alphabet size is at most equal to the input alphabet size: $|\mathcal{Y}| \leq |\mathcal{X}|$.*

The optimal solution is an extremal mechanism, since the absolute value of the log-likelihood ratios can only take one of the two extremal values (see Figure 1). We refer to such a mechanism as a staircase mechanism, and define the *family of staircase mechanisms* as

$$\mathcal{S}_\varepsilon \equiv \{Q \,|\, \text{satisfying (7)}\} .$$

This family includes all the optimal mechanisms (for all choices of $\varepsilon \geq 0$, $P_0$, $P_1$ and $f$), and since (7) implies (6), staircase mechanisms are locally differentially private.

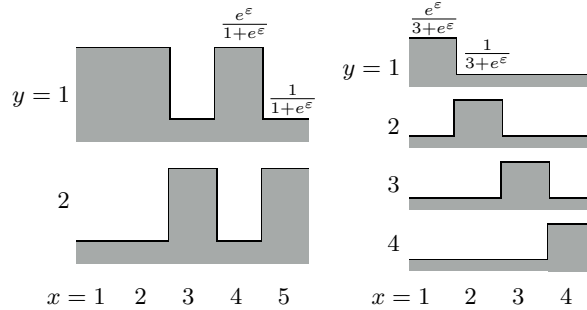

Figure 1: Examples of staircase mechanisms: the binary and randomized response mechanisms.

For global differential privacy, we can generalize the definition of staircase mechanisms to hold for all neighboring database queries $x, x'$ (or equivalently within some sensitivity), and recover all known existing optimal mechanisms. Precisely, the geometric mechanism shown to be optimal in [17], and the mechanisms shown to be optimal in [14, 16] (also called staircase mechanisms) are special cases of the staircase mechanisms defined above. We believe that the characterization of these extremal mechanisms and the analysis techniques developed in this paper can be of independent interest to researchers interested in optimal mechanisms for global privacy and more general utilities.

**Combinatorial representation of the staircase mechanisms.** Now that we know staircase mechanisms are optimal, we can try to combinatorially search for the best staircase mechanism for any fixed $\varepsilon$, $P_0$, $P_1$, and $f$. To this end, we give a simple representation of all staircase mechanisms, exploiting the fact that they are scaled copies of a finite number of patterns.

Let $Q \in \mathbb{R}^{|\mathcal{X}| \times |\mathcal{Y}|}$ be a staircase mechanism and $k = |\mathcal{X}|$ denote the input alphabet size. Then, using the definition of staircase mechanisms, $Q(y|x)/Q(y|x') \in \{e^{-\varepsilon}, 1, e^\varepsilon\}$ and each column $Q(y|\cdot)$ must be proportional to one of the canonical staircase patterns. For example, when $k = 3$,

there are $2^k = 8$ canonical patterns. Define a staircase pattern matrix $S^{(k)} \in \{1, e^{\varepsilon}\}^{k \times (2^k)}$ taking values either 1 or $e^{\varepsilon}$, such that the $i$-th column of $S^{(k)}$ has a staircase *pattern* corresponding to the binary representation of $i - 1 \in \{0, \ldots, 2^k - 1\}$. We order the columns of $S^{(k)}$ in this fashion for notational convenience. For example,

$$S^{(3)} = \begin{bmatrix} 1 & 1 & 1 & 1 & e^{\varepsilon} & e^{\varepsilon} & e^{\varepsilon} & e^{\varepsilon} \\ 1 & 1 & e^{\varepsilon} & e^{\varepsilon} & 1 & 1 & e^{\varepsilon} & e^{\varepsilon} \\ 1 & e^{\varepsilon} & 1 & e^{\varepsilon} & 1 & e^{\varepsilon} & 1 & e^{\varepsilon} \end{bmatrix} .$$

For all values of $k$, there are exactly $2^k$ such patterns, and any column of $Q(y|\cdot)$ is a scaled version of one of the columns of $S^{(k)}$. Using this "pattern" matrix, we will show that we can represent (an equivalence class of) any staircase mechanism $Q$ as

$$Q = S^{(k)}\Theta , \tag{8}$$

where $\Theta \in \mathbb{R}^{2^k \times 2^k}$ is a diagonal matrix representing the scaling of the columns of $S^{(k)}$. We can now formulate the problem of maximizing the divergence between the induced marginals as a linear program and prove that it is equivalent the original nonlinear program.

**Theorem 2.2.** *For any $\varepsilon$, any pair of distributions $P_0$ and $P_1$, and any $f$-divergence, the nonlinear program of* (5) *and the following linear program have the same optimal value*

$$\begin{aligned} \underset{\Theta \in \mathbb{R}^{2^k \times 2^k}}{maximize} \quad & \sum_{i=1}^{2^k} \mu(S_i^{(k)})\Theta_{ii} && (9) \\ subject\ to \quad & S^{(k)}\Theta \mathbb{1} = \mathbb{1} , \\ & \Theta \text{ is a diagonal matrix} , \\ & \Theta \geq 0 , \end{aligned}$$

*where $\mu(S_i^{(k)}) = (\sum_{x \in \mathcal{X}} P_1(x) S_{xi}^{(k)}) f(\sum_{x \in \mathcal{X}} P_0(x) S_{xi}^{(k)} / \sum_{x \in \mathcal{X}} P_1(x) S_{xi}^{(k)})$ and $S_i^{(k)}$ is the $i$-th column of $S^{(k)}$, such that $D_f(M_0||M_1) = \sum_{i=1}^{2^k} \mu(S_i^{(k)})\Theta_{ii}$. The solutions of* (5) *and* (9) *are related by* (8).

The infinite dimensional nonlinear program of (5) is now reduced to a finite dimensional linear program. The first constraint ensures that we get a valid probability transition matrix $Q = S^{(k)}\Theta$ with a row sum of one. One could potentially solve this LP with $2^k$ variables but its computational complexity scales exponentially in the alphabet size $k = |\mathcal{X}|$. For practical values of $k$ this might not always be possible. However, in the following section, we give a precise description for the optimal mechanisms in the high privacy and low privacy regimes.

In order to understand the above theorem, observe that both the $f$-divergences and the differential privacy constraints are invariant under *permutation* (or relabelling) of the columns of a privatization mechanism $Q$. For example, the KL-divergence $D_{kl}(M_0||M_1)$ does not change if we permute the columns of $Q$. Similarly, both the $f$-divergences and the differential privacy constraints are invariant under *merging/splitting* of outputs with the same pattern. To be specific, consider a privatization mechanism $Q$ and suppose there exist two outputs $y$ and $y'$ that have the same pattern, i.e. $Q(y|\cdot) = C\,Q(y'|\cdot)$ for some positive constant $C$. Then, we can consider a new mechanism $Q'$ by merging the two columns corresponding to $y$ and $y'$. Let $y''$ denote this new output. It follows that $Q'$ satisfies the differential privacy constraints and the resulting $f$-divergence is also preserved. Precisely, using the fact that $Q(y|\cdot) = C\,Q(y'|\cdot)$, it follows that

$$\frac{M_0'(y'')}{M_1'(y'')} = \frac{\sum_x (Q(y|x) + Q(y'|x))P_0(x)}{\sum_x (Q(y|x) + Q(y'|x))P_1(x)} = \frac{(1+C)\sum_x Q(y|x)P_0(x)}{(1+C)\sum_x Q(y|x)P_1(x)} = \frac{M_0(y)}{M_1(y)} = \frac{M_0(y')}{M_1(y')} ,$$

and thus the corresponding $f$-divergence is invariant:

$$f\Big(\frac{M_0(y)}{M_1(y)}\Big)M_1(y) + f\Big(\frac{M_0(y')}{M_1(y')}\Big)M_1(y') = f\Big(\frac{M_0'(y'')}{M_1'(y'')}\Big)M_1'(y'') .$$

We can naturally define equivalence classes for staircase mechanisms that are equivalent up to a permutation of columns and merging/splitting of columns with the same pattern:

$$[Q] = \{Q' \in \mathcal{S}_{\varepsilon} \,|\, \text{exists a sequence of permutations and merge/split of columns from } Q' \text{ to } Q\} . \tag{10}$$

To represent an equivalence class, we use a mechanism in $[Q]$ that is ordered and merged to match the patterns of the pattern matrix $S^{(k)}$. For any staircase mechanism $Q$, there exists a possibly different staircase mechanism $Q' \in [Q]$ such that $Q' = S^{(k)}\Theta$ for some diagonal matrix $\Theta$ with nonnegative entries. Therefore, to solve optimization problems of the form (5), we can restrict our attention to such representatives of equivalent classes. Further, for privatization mechanisms of the form $Q = S^{(k)}\Theta$, the $f$-divergences take the form given in (9), a simple linear function of $\Theta$.

## 2.2 Optimal mechanisms in high and low privacy regimes

For a given $P_0$ and $P_1$, the *binary mechanism* is defined as a staircase mechanism with only two outputs $y \in \{0, 1\}$ satisfying (see Figure 1)

$$Q(0|x) = \begin{cases} \frac{e^\varepsilon}{1+e^\varepsilon} & \text{if } P_0(x) \geq P_1(x) , \\ \frac{1}{1+e^\varepsilon} & \text{if } P_0(x) < P_1(x) . \end{cases} \quad Q(1|x) = \begin{cases} \frac{e^\varepsilon}{1+e^\varepsilon} & \text{if } P_0(x) < P_1(x) , \\ \frac{1}{1+e^\varepsilon} & \text{if } P_0(x) \geq P_1(x) . \end{cases} \quad (11)$$

Although this mechanism is extremely simple, perhaps surprisingly, we will establish that this is the optimal mechanism when high level of privacy is required. Intuitively, the output is very noisy in the high privacy regime, and we are better off sending just one bit of information that tells you whether your data is more likely to have come from $P_0$ or $P_1$.

**Theorem 2.3.** *For any pair of distributions $P_0$ and $P_1$, there exists a positive $\varepsilon^*$ that depends on $P_0$ and $P_1$ such that for any $f$-divergences and any positive $\varepsilon \leq \varepsilon^*$, the binary mechanism maximizes the $f$-divergence between the induced marginals over all $\varepsilon$-local differentially private mechanisms.*

This implies that in the high privacy regime, which is a typical setting studied in much of differential privacy literature, the binary mechanism is a universally optimal solution for all $f$-divergences in (5). In particular this threshold $\varepsilon^*$ is *universal*, in that it does not depend on the particular choice of which $f$-divergence we are maximizing. This is established by proving a very strong statistical dominance using Blackwell's celebrated result on comparisons of statistical experiments [4]. In a nutshell, we prove that for sufficiently small $\varepsilon$, the output of any $\varepsilon$-locally differentially private mechanism can be simulated from the output of the binary mechanism. Hence, the binary mechanism dominates over all other mechanisms and at the same time achieves the maximum divergence. A similar idea has been used previously in [27] to exactly characterize how much privacy degrades under composition.

The optimality of binary mechanisms is not just for high privacy regimes. The next theorem shows that it is *the* optimal solution of (5) for all $\varepsilon$, when the objective function is the total variation $D_f(M_0\|M_1) = \|M_0 - M_1\|_{\text{TV}}$.

**Theorem 2.4.** *For any pair of distributions $P_0$ and $P_1$, and any $\varepsilon \geq 0$, the binary mechanism maximizes total variation between the induced marginals $M_0$ and $M_1$ among all $\varepsilon$-local differentially private mechanisms.*

When maximizing the KL-divergence between the induced marginals, we show that the binary mechanism still achieves a good performance for all $\varepsilon \leq C$ where $C \geq \varepsilon^*$ now does not depend on $P_0$ and $P_1$. For the special case of KL-divergence, let OPT denote the maximum value of (5) and BIN denote the KL-divergence when the binary mechanism is used. The next theorem shows that

$$\text{BIN} \geq \frac{1}{2(e^\varepsilon + 1)^2}\text{OPT} .$$

**Theorem 2.5.** *For any $\varepsilon$ and for any pair of distributions $P_0$ and $P_1$, the binary mechanism is an $1/(2(e^\varepsilon + 1)^2)$ approximation of the maximum KL-divergence between the induced marginals $M_0$ and $M_1$ among all $\varepsilon$-locally differentially private mechanisms.*

Note that $2(e^\varepsilon + 1)^2 \leq 32$ for $\varepsilon \leq 1$, and $\varepsilon \leq 1$ is a common regime of interest in differential privacy. Therefore, we can always use the simple binary mechanism in this regime and the resulting divergence is at most a constant factor away from the optimal one.

The *randomized response mechanism* is defined as a staircase mechanism with the same set of outputs as the input, $\mathcal{Y} = \mathcal{X}$, satisfying (see Figure 1)

$$Q(y|x) = \begin{cases} \frac{e^\varepsilon}{|\mathcal{X}|-1+e^\varepsilon} & \text{if } y = x , \\ \frac{1}{|\mathcal{X}|-1+e^\varepsilon} & \text{if } y \neq x . \end{cases}$$

It is a randomization over the same alphabet where we are more likely to give an honest response. We view it as a multiple choice generalization of the randomized response proposed by Warner [29], assuming equal privacy level for all choices. We establish that this is the optimal mechanism when low level of privacy is required. Intuitively, the noise is small in the low privacy regime, and we want to send as much information about our current data as allowed, but no more. For a special case of maximizing KL-divergence, we show that the *randomized response mechanism* is the optimal solution of (5) in the low privacy regime ($\varepsilon \geq \varepsilon^*$).

**Theorem 2.6.** *There exists a positive $\varepsilon^*$ that depends on $P_0$ and $P_1$ such that for any $P_0$ and $P_1$, and all $\varepsilon \geq \varepsilon^*$, the randomized response mechanism maximizes the KL-divergence between the induced marginals over all $\varepsilon$-locally differentially private mechanisms.*

## 2.3 Lower bounds in differential privacy

In this section, we provide converse results on the fundamental limit of differentially private mechanisms. These results follow from our main theorems and are of independent interest in other applications where lower bounds in statistical analysis are studied [3, 21, 6, 9]. For example, a bound similar to (12) was used to provide converse results on the sample complexity for statistical estimation with differentially private data in [10].

**Corollary 2.7.** *For any $\varepsilon \geq 0$, let $Q$ be any conditional distribution that guarantees $\varepsilon$-local differential privacy. Then, for any pair of distributions $P_0$ and $P_1$, and any positive $\delta > 0$, there exists a positive $\varepsilon^*$ that depends on $P_0$, $P_1$, and $\delta$ such that for any $\varepsilon \leq \varepsilon^*$, the induced marginals $M_0$ and $M_1$ satisfy the bound*

$$D_{\mathrm{kl}}\big(M_0||M_1\big) + D_{\mathrm{kl}}\big(M_1||M_0\big) \quad \leq \quad \frac{2(1+\delta)(e^\varepsilon - 1)^2}{(e^\varepsilon + 1)} \left\| P_0 - P_1 \right\|_{\mathrm{TV}}^2 . \tag{12}$$

This follows from Theorem 2.3 and the fact that under the binary mechanism, $D_{\mathrm{kl}}\big(M_0||M_1\big) = \left\| P_0 - P_1 \right\|_{\mathrm{TV}}^2 (e^\varepsilon - 1)^2 /(e^\varepsilon + 1) + O(\varepsilon^3)$ . Compared to [10, Theorem 1], we recover their bound of $4(e^\varepsilon - 1)^2 \left\| P_0 - P_1 \right\|_{\mathrm{TV}}^2$ with a smaller constant. We want to note that Duchi et al.'s bound holds for all values of $\varepsilon$ and uses different techniques. However no achievable mechanism is provided. We instead provide an explicit mechanism that is optimal in high privacy regime.

Similarly, in the high privacy regime, we can show the following converse result.

**Corollary 2.8.** *For any $\varepsilon \geq 0$, let $Q$ be any conditional distribution that guarantees $\varepsilon$-local differential privacy. Then, for any pair of distributions $P_0$ and $P_1$, and any positive $\delta > 0$, there exists a positive $\varepsilon^*$ that depends on $P_0$, $P_1$, and $\delta$ such that for any $\varepsilon \geq \varepsilon^*$, the induced marginals $M_0$ and $M_1$ satisfy the bound*

$$D_{\mathrm{kl}}\big(M_0||M_1\big) + D_{\mathrm{kl}}\big(M_1||M_0\big) \quad \leq \quad D_{\mathrm{kl}}(P_0||P_1) - (1 - \delta)G(P_0, P_1)e^{-\varepsilon} .$$

*where $G(P_0, P_1) = \sum_{x \in \mathcal{X}}(1 - P_0(x)) \log(P_1(x)/P_0(x))$.*

This follows directly from Theorem 2.6 and the fact that under the randomized response mechanism, $D_{\mathrm{kl}}(M_0||M_1) = D_{\mathrm{kl}}(P_0||P_1) - G(P_0, P_1)e^{-\varepsilon} + O(e^{-2\varepsilon})$ .

Similarly for total variation, we can get the following converse result. This follows from Theorem 2.4 and explicitly computing the total variation achieved by the binary mechanism.

**Corollary 2.9.** *For any $\varepsilon \geq 0$, let $Q$ be any conditional distribution that guarantees $\varepsilon$-local differential privacy. Then, for any pair of distributions $P_0$ and $P_1$, the induced marginals $M_0$ and $M_1$ satisfy the bound $\left\| M_0 - M_1 \right\|_{\mathrm{TV}} \leq ((e^\varepsilon - 1)/(e^\varepsilon + 1)) \left\| P_0 - P_1 \right\|_{\mathrm{TV}}$ , and equality is achieved by the binary mechanism.*

## 2.4 Connections to hypothesis testing

Under the data collection scenario, there are $n$ individuals each with data $X_i$ sampled from a distribution $P_\nu$ for a fixed $\nu \in \{0, 1\}$. Let $Q$ be a non-interactive privatization mechanism guaranteeing $\varepsilon$-local differential privacy. The privatized views $\{Y_i\}_{i=1}^n$, are independently distributed according to one of the induced marginals $M_0$ or $M_1$ defined in (2).

Given the privatized views $\{Y_i\}_{i=1}^n$, the data analyst wants to test whether they were generated from $M_0$ or $M_1$. Let the null hypothesis be $H_0 : Y_i$'s are generated from $M_0$, and the alternative hypothesis $H_1 : Y_i$'s are generated from $M_1$. For a choice of rejection region $R \subseteq \mathcal{Y}^n$, the probability of false alarm (type I error) is $\alpha = M_0^n(R)$ and the probability of miss detection (type II error) is $\beta = M_1^n(\mathcal{Y}^n \setminus R)$. Let $\beta^\delta = \min_{R \subseteq \mathcal{Y}^n, \alpha < \alpha^*} \beta$ denote the minimum type II error achievable while keeping type I error rate at most $\alpha^*$. According to Chernoff-Stein lemma, we know that

$$\lim_{n \to \infty} \frac{1}{n} \log \beta^{\alpha^*} = -D_{\mathrm{kl}}(M_0 \| M_1) \ .$$

Suppose the analyst knows $P_0$, $P_1$, and $Q$. Then, in order to achieve optimal asymptotic error rate, one would want to maximize the KL-divergence between the induced marginals over all $\varepsilon$-locally differentially private mechanisms $Q$. Theorems 2.3 and 2.6 provide an explicit construction of the optimal mechanisms in high and low privacy regimes. Further, our converse results in Section 2.3 provides a fundamental limit on the achievable error rates under differential privacy. Precisely, with data collected from an $\varepsilon$-locally differentially privatization mechanism, one cannot achieve an asymptotic type II error smaller than

$$\lim_{n \to \infty} \frac{1}{n} \log \beta^{\alpha^*} \geq -\frac{(1+\delta)(e^\varepsilon - 1)^2}{(e^\varepsilon + 1)} \|P_0 - P_1\|_{\mathrm{TV}}^2 \ \geq -\frac{(1+\delta)(e^\varepsilon - 1)^2}{2(e^\varepsilon + 1)} D_{\mathrm{kl}}(P_0 \| P_1) \ ,$$

whenever $\varepsilon \leq \varepsilon^*$, where $\varepsilon^*$ is dictated by Theorem 2.3. In the equation above, the second inequality follows from Pinsker's inequality. Since $(e^\varepsilon - 1)^2 = O(\varepsilon^2)$ for small $\varepsilon$, the effective sample size is now reduced from $n$ to $\varepsilon^2 n$. This is the price of privacy. In the low privacy regime where $\varepsilon \geq \varepsilon^*$, for $\varepsilon^*$ dictated by Theorem 2.6, one cannot achieve an asymptotic type II error smaller than

$$\lim_{n \to \infty} \frac{1}{n} \log \beta^{\alpha^*} \geq -D_{\mathrm{kl}}(P_0 \| P_1) + (1 - \delta) G(P_0, P_1) e^{-\varepsilon} \ .$$

## 3  Discussion

In this paper, we have considered $f$-divergence utility functions and assumed a setting where individuals cannot collaborate (communicate with each other) before releasing their data. It turns out that the optimality results presented in Section 2 are general and hold for a large class of convex utility function [22]. In addition, the techniques developed in this work can be generalized to find optimal privatization mechanisms in a setting where different individuals can collaborate interactively and each individual can be an analyst [23].

Binary hypothesis testing is a canonical statistical inference problem with a wide range of applications. However, there are a number of nontrivial and interesting extensions to our work. Firstly, in some scenarios the $X_i$'s could be correlated (e.g., when different individuals observe different functions of the same random variable). In this case, the data analyst is interested in inferring whether the data was generated from $P_0^n$ or $P_1^n$, where $P_\nu^n$ is one of two possible joint priors on $X_1, ..., X_n$. This is a challenging problem because knowing $X_i$ reveals information about $X_j$, $j \neq i$. Therefore, the utility maximization problems for different individuals are coupled in this setting. Secondly, in some cases the data analyst need not have access to $P_0$ and $P_1$, but rather two classes of prior distribution $P_{\theta_0}$ and $P_{\theta_1}$ for $\theta_0 \in \Lambda_0$ and $\theta_1 \in \Lambda_1$. Such problems are studied under the rubric of universal hypothesis testing and robust hypothesis testing. One possible direction is to select the privatization mechanism that maximizes the worst case utility: $Q^* = \arg\max_{Q \in \mathcal{D}_\varepsilon} \min_{\theta_0 \in \Lambda_0, \theta_1 \in \Lambda_1} D_f(M_{\theta_0} \| M_{\theta_1})$, where $M_{\theta_\nu}$ is the induced marginal under $P_{\theta_\nu}$. Finally, the more general problem of private $m$-ary hypothesis testing is also an interesting but challenging one. In this setting, the $X_i$'s can follow one of $m$ distributions $P_0, P_1, ..., P_{m-1}$, and therefore the $Y_i$'s can follow one of $m$ distributions $M_0, M_1, ..., M_{m-1}$. The utility can be defined as the average $f$-divergence between any two distributions: $1/(m(m-1)) \sum_{i \neq j} D_f(M_i \| M_j)$, or the worst case one: $\min_{i \neq j} D_f(M_i \| M_j)$.

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
