[Reviews · NeurIPS 2014]

Submitted by Assigned_Reviewer_3

Paper ID: 1486
Title: Extremal Mechanism for Local Differential Privacy

NOTE: Due to the short reviewing time and out of fairness to the other papers I was reviewing, I DID NOT do more than glance over the supplementary material. In most cases, I therefore did not verify that results claimed by the authors are correct, but instead checked that they are plausible.

Summary: The authors study optimal mechanisms for local differential privacy. The problem they address is one in which a statistician seeks to test whether the data comes from a distribution $P_0$ or $P_1$ on the basis of a locally private version of the data. One way to measure the difficulty of this decision is by computing an $f$-divergence. The authors show that optimal mechanisms follow a "staircase" pattern and that the output alphabet for a sanitization mechanism need only be as large as the input alphabet. They further show that a particular 1-bit mechanism (which outputs a single bit per input datum) is optimal as $\epsilon \to 0$ and provide approximation ratios for total variation and KL divergence.

Quality, clarity, originality and significance: This is a very nice theoretical result on the structure of differentially private solutions, especially for statistical decision problems. The paper is clear from the theory perspective, and provides an original take on mechanism design for differential privacy.

Comments: (comments are numbered to allow for easy reference in the rebuttal)

1) It's not clear from the introduction what the authors mean by the "multi-party setting." Do each of the parties have data? Is each an analyst? I might remove these comments without further clarification.

2) The randomized response mechanism here is actually different than the mechanism of Duchi et al., which binarizes a multinomial variable. In the author's language, this models a variable X from a finite alphabet |\mathcal{X}| as a binary indicator in $\{ 0,1 \}^{|\mathcal{X}|}$, and then forms $Y$ also in $\{ 0,1 \}^{|\mathcal{X}|}$ "by sampling... coordinates independently." Thus the alphabet size for $\mathcal{Y}$ is significantly larger than $|\mathcal{X}|$. Some commentary on the difference between the author's mechanism and the other randomized response mechanism may be warranted.

3) The authors claim that $\epsilon < 1$ is the interesting case, but in some applications choosing $\epsilon < 1$ is not possible due to smaller sample size, high dimension, or other factors. That is, $\epsilon < 1$ may result in very poor utility.

4) The authors should probably say what $\epsilon^*$ does depend on (the alphabet size? The distributions $P_0$ and $P_1$?)

5) One thing that might not be clear to the more casual reader (who is aware of more algorithmic work on privacy) is why the problem restriction to $P_0$ and $P_1$ is particularly relevant. Because the authors consider local privacy, it may take a little more hand-holding to get readers to appreciate this point, especially because differential privacy is often explained in terms of neighboring databases where only 1 individual is different. Naively, one may think that as long as $P_0 \ne P_1$, a sample $X^n \sim P_0$ should differ significantly from a sample $X^n \sim P_1$ (i.e., in many more than 1 individual).

6) One point that was not quite clear is whether the mechanism itself will depend on utility measure, or if there are mechanisms which are simultaneously optimal for several mechanisms. Related to this is whether the output alphabet $\mathcal{Y}$ depends on the $f$ for the divergence.

7) How would the solution change if the mechanism designer did not have a pair of distributions but instead two classes of distributions? Would the same approach work?

Please summarize your review in 1-2 sentences: This paper studies the structure of optimal mechanisms under local differential privacy when the utility measure is an $f$-divergence between two data distributions. The results are an important contribution to the literature on privacy and statistics.

Quality Score (1-10): 9

Impact Score (1-2): 1

Confidence (1-5): 5

EDITED AFTER AUTHOR FEEDBACK: The authors spent most of their time addressing concerns from a different reviewer. However, in the discussion I came to agree with that reviewer that the paper is not well-written for the non expert. Therefore I would strongly encourage the author to make the paper a bit more friendly to the casual NIPS reader (perhaps at the expense of moving more technical details to the appendix).
Summary: This paper studies the structure of optimal mechanisms under local differential privacy when the utility measure is an $f$-divergence between two data distributions. The results are an important contribution to the literature on privacy and statistics.

Submitted by Assigned_Reviewer_6

The paper discusses new mechanisms for differential privacy. In particular the paper suggests a very nice staircase construction providing an optimal solution for the problem of maximizing the distance between two hypotheses while maintaining differential privacy.
I am new to the concept of the differential privacy and thus was reading the paper as a non-expert interested to learn about the subject. The paper is written clear enough for me to understand the setting and then to appreciate suggested solution. The essence of the result is in replacing the inequality (local diff. privacy) condition (7) by its discrete analog (8) and then constructing the solution (maximizing difference between the two hypotheses) explicitly via the staircase mechanism. Next, the remaining degree of freedom (within the family of staircase mechanisms) is parameterized in a continuous way, by Eq.(9), and it is shown that the original nonlinear optimization is replaced by an LP given by (10). Unfortunately the LP is still exponential in the input alphabet size.
I like the paper and recommend acceptance. I would also like the authors to take care of the following minor problems:
1) \varepsilon is mentioned in the abstract where it was not yet defined. Rewrite the sentence stating it in words.
2) You start from a general multi-client case and then transition to the client-server setting. My understanding is that the statistical independence of the clients is the key for the number of clients not to enter in your further consideration. However, it would also be useful to comment on how to deal with the more general (correlated) case. Does your method offers a hint on that?
3) Your final LP formulation is exponential in the input alphabet size. Do you envision a relaxation which reduces the complexity? If you are, I suggest adding a brief comment.
Summary: The theoretical paper reports interesting results. I do recommend acceptance.

Submitted by Assigned_Reviewer_13

The paper claims to make a contribution to local differential privacy. Unfortunately, I didn't sufficiently understand the text to confidently interprete and summarize the content.

The quality of the paper was not sufficient to let me understand its main message. In particular, the writing is quite vague and unclear.

The paper describes some related work, but not being able to interprete the fine details of the contribution, I couldn't assess the originality and significance.

The following are detailed comments in order of occurrence in the text:

The abstract is a bit vague wrt. the topic and contribution of the paper. It says "We address a general problem of utility maximization under local differential privacy." but doesn't specify what this "general problem of utility maximization" is. The rest of the abstract is therefore over the top of my head upon first reading.

The authors define local privacy as when "individuals disclose their personal information (e.g. voluntarily on social network sites).", but if individuals disclose data to the world, what is then the problem of privacy?

The authors discuss that "differential privacy has been extended to local privacy" and use in their formal definition "privatized views Y_i" without explaining what are privatized views (remember from the previous comment that the data is already public). These "view" (usually inputs to the viewer) are also called "outputs" in the next sentence. In this definition, it is also not clear why exactly two distributions (P_0 and P_1) are considered. The sentence "We consider Q’s satisfying the privacy conditions," is unclear in not specifying what is the "privacy condition" (is it the differential privacy discussed later in the definition?). The formula of the definition looks a common one, so I guess the problem is mainly in the formalization and wording.

This definition of differential privacy uses Q for a conditional probability distribution, where the conditions are on one single x_i and n-1 different y_j's. Formula (2) uses Q with conditioning on n x_i's. The number of conditionings is the same, but it is unclear how they map to each other. Moreover, the "privatization" formulation suggests that the x_i and y_j are not necessarily elements of the same domain.

The introduction further states that the power to discriminate data from P_0 from data from P_1 depends on the marginals M_\nu. However, it seems there are other ways to discriminate, e.g. by just considering the distribution of the values of X_i for some i (everyone knows at least one X_i).

Line 110: "Q the minimized the error" -> "Q that minimizes the error"

It is unclear why Equation (4) introduces f-divergence and then (5) specializes to KL-divergence (which the paper says is a special case).

Line 175: and marginal of the privatized output -> ungrammatical

Section 2 starts saying "We first give a formal definition of the staircase mechanisms," but I don't find such formal definition before theorem 2.1. Theorem 2.1 is not proved in the main paper and does not refer to the supplementary material. The proof in the supplementary material seems to use staircase notion (which as I said don't seem to be provided before that point).

After this point, the technical content gets increasingly confusing.

The paper has an usual structure, has no conclusion section, does not outline the structure at the end of the introduction.

The paper does not provide experimental results, examples, illustrations, figures, or other attempts at making the content somewhat easier to digest.

-------------------------------------------

Additional comments after the author's rebuttal.

W.r.t. Comment 3, Authors say: "the authors say: We provide in lines 121-157, a summary of our contributions in the context of existing literature."
I reread now this part, and try to comment based on what I understand from the paper, the discussion with other reviewers and the author's response:
* Line 122: In fact, the authors are not solving an optimization problem under local differential privacy constraints, but they are optimizing privatization mechanisms under local differential privacy constraints. Taking the sentence literally, one (of the several) possible semantics for this sentence in isolation would be that we are given a privatized view and the data analyst should try to find the best possible value for some (model) parameter. The second part of the sentence (resulting in a family of optimal mechanisms) is inconsistent with the first part.
* Terms such as "large class", "family", "important", ... are rather vague, and will need to be instantiated to the concrete contribution at some point (which does not happen in the part 121-157). It is (especially here) unclear why "optimization problem" gets the specification "large class of", as the only optimization problem the authors are solving is Equation (5), while "large class" strengthens the wrong semantics that it concerns a (not yet explained) other class of optimization problems. I guess that "large class" does not refer to optimization problems but to a large class of probability distributions P_\nu, large classes of privatization schemes Q, etc.
* Line 124: Next, the reader stumbles upon "the differential privacy area, where the basic mechanisms have been few". What is a basic mechanism? It could refer to privatization mechanisms, but why are they called basic (rather than privatization) then? A much simpler and less confusing formulation would be "... to the differential privacy area where allmost no results are known on the optimality of privatization mechanisms." (if at least this is what the authors mean).
* Somehow, the reader also needs to realize that the "extremal" from the title is a synonym for the "optimal" in this paragraph. Avoiding the use of synonyms often avoids confusion.
* Line 125 mentions a linear program over "the alphabet |\mathcal{X}|. Probably the "alphabet size" is meant (|.| gives a number). By the way, in line 64, \mathcal{X} was not introduced (and the reader learns here that it is called "alphabet"). Up to here, the reader may think that \mathcal{X} is an uncountably infinite set (e.g. vector space). This impression is strengthened by the usage of integrals in the formulas up to here.
* Line 126-128 seems to revoke the original stronger claim that optimal mechanisms can be found using this paper. It now seems to hard to compute, and the paper will only give optimal mechanisms in certain "regimes", which usually means something like "the optimality result holds for the extremes (e.g. 0 and infinity), and the more you are in the middle between these extremes, the large is the suboptimality.". Still, later in lines 144-146, the text compares with related work and says that this other work only provides "near-optimality" while this paper does better.
* line 129 announces bounds on differential privacy: Suddenly, it is not local anymore. What exactly is lower bounded? In terms of what is the lower bound given? The statement is quite vague.
* In "recovering some known results.", what is "recover"? It sounds as this "recovering" is not giving new results, but repeating what has been published earlier? Or do you mean that you adapt existing bounds to a new setting where they originally did not yet apply?
* In Line 129, The sentence "As a motivating example, we provide optimal mechanisms" is again confusing. I thought that already line 123 announced optimal mechanisms? So are the optimal mechanisms a motivating example for the motivating mechanisms? What is the difference?
* Ine line 134, it is unclear what "a converse to the optimization problem" means, the following phrase (and show ...) does not disambiguate this completely.

In conclusion, even though the authors indeed provide in the mentioned lines a summary of their contributions, it is vague, and the text provides many opportunities to the average reader (who does not have author responses and fellow reviewers) for getting confused. In allmost all cases, simple reformulations exist which make the text clearer, more precise, less ambiguous and less confusing.

W.r.t. Comment 6: Authors say: "The specific quoted phrase is embedded in a larger sentence. Reading the entire sentence, local privacy DOES NOT mean that individuals reveal their personal information, but instead local privacy is required IN THE CONTEXT where individuals reveal their personal information. Local differential privacy is when users first privatize their personal data (X_i's) and then release privatized views (Y_i's) to a data collector. Therefore, the data collector never observes the X_i's but instead has access to the Y_i's."
It indeed often happens that the authors write sentences which have certain grammatical features which do not allow an interpretation consistent with other sentences or parts of sentences. In this case, the quoted phrase IS NOT embedded in a larger sentence which may let the reader suspect he is being misled. In particular, the complete sentence is "The need for data privacy appears in two different contexts: one in the data collection scenario, as in when individuals disclose their personal information (e.g. voluntarily on social network sites).". Later, the paper says that this first scenario is local differential privacy. It is true that there are other sentences in the paper which describe local differential privacy better, but why then first give a wrong description?
My specific comment refers amongst others to the fact that the example is clearly not a good one. Users providing their information on a social network site do not privatize their information in some complicated way. They do not know the (one or more) data analysts who will analyze the data, and hence even if they would apply a privatization scheme, they would not be able to use the content in this paper to select an optimal privatization scheme.

Even if we ignore the specific ill-chosen example, it can't be denied that the paper assumes that there is only one data analyst willing to test only one hypothesis. In practice, this is not realistic. Data is published, potentially in some privatized way, but then usually a data analyst will have more then a binary question.

Authors say: "7A) Local differential privacy has been studied in the literature. Our notations, definitions, and problem statement have been chosen to be consistent with recent prior papers on the subject (which we cite in our paper). Specifically, the terms "privatized views", "privatized outputs", and "privatization mechanism" are chosen to be consistent with their use in [10]."
Ideally, a paper is self-contained as should not require from the reader that he digs into a lot of other literature to grasp the main message.
Even if the authors choose to not define terms already used in the literature, why then let the reader guess for a long time and not simply start by saying that the concepts and notations in this paper are defined formally in [10] ?
More generally, several boosting sentences such as "We first give a formal definition ..." let the reader expect a certain level of rigorousness, and cause the reader to think his missed something in this paper while he is not able to find it. If you don't plan to give a formal definition, or if you give it in supplementary materials, ... my advice is to simply tell this to the reader.

Still, it is my belief (now understanding things better), that there are simple ways to convey most of this essential information to the reader without using significantly more space. Even if a clearer explanation would require more space, I feel it is worth moving one or two technical derivations to the supplementary materials.

Authors say: "7B) Binary hypothesis testing is a canonical statistical inference problem with a wide range of applications. As is evident from [10] and this paper, understanding the fundamental limits to private binary hypothesis testing is a challenging problem. The more general problem of private M-ary hypothesis testing is also interesting, but is outside of the scope of this paper."
I agree with that. But then, this is an interesting point of discussion and possible future work. Why not have a short "conclusion and future directions" section where you outline the limitations of the current paper and remaining open questions?

w.r.t. comment 16, I feel that examples, discussions and illustrations don't necessarily require figures. Well-written sentences often can help a lot.

Summary: For me, the writing of this paper is insufficiently clear. The paper has an introduction containing already a lot of technical stuff which is not well explained, and a section 2 providing technical material. There are no conclusions, discussions, examples, illustrations, experiments or other efforts to help the reader understand the content.
Author Feedback
Author rebuttal: Reviewer_13:

Our responses are enumerated based on the line breaks in the review. We write "Blank" whenever the comment does not require a response.

1) Blank

2) Blank

3) We provide in lines 121-157, a summary of our contributions in the context of existing literature.

4) Blank

5) As indicated in the abstract, there is a fundamental trade-off between privacy and utility. This trade-off is posed as "a general problem of utility maximization under local differential privacy".

6) The specific quoted phrase is embedded in a larger sentence. Reading the entire sentence, local privacy DOES NOT mean that individuals reveal their personal information, but instead local privacy is required IN THE CONTEXT where individuals reveal their personal information. Local differential privacy is when users first privatize their personal data (X_i's) and then release privatized views (Y_i's) to a data collector. Therefore, the data collector never observes the X_i's but instead has access to the Y_i's.

7A) Local differential privacy has been studied in the literature. Our notations, definitions, and problem statement have been chosen to be consistent with recent prior papers on the subject (which we cite in our paper). Specifically, the terms "privatized views", "privatized outputs", and "privatization mechanism" are chosen to be consistent with their use in [10].

7B) Binary hypothesis testing is a canonical statistical inference problem with a wide range of applications. As is evident from [10] and this paper, understanding the fundamental limits to private binary hypothesis testing is a challenging problem. The more general problem of private M-ary hypothesis testing is also interesting, but is outside of the scope of this paper.

7C) In lines 68-69, we define Q as a privatization mechanism that obeys appropriate privacy conditions. Local differential privacy is one such way of defining the privacy conditions (and is the main focus of the entire paper) and is formally defined in the following lines, in Equation 1 (line 69-75).

8) Our notations in Eqs. 1-3 are standard in the literature and is consistent with recent work [10]. The Q in Equation 3 is a special case of the Q in Equation 1. The Q in Equation 1 allows for interaction (multi-party setting) whereas the one in Equation 3 does not (client-server setting). Q^n in Equation 2 is defined as the joint conditional distribution of Y_1,.,Y_n, (this might resolve the confusion in terms of the difference in domains of Q and Q^n in X_i's and Y_i's). The choice of notation mirrors that in Equation 1 on purpose, since in the important non-interactive setting Q^n is the n-fold convolution of Q.

9) As mentioned in item (6) above, the X_i's are not released and only the Y_i's are released. The Y_i's are distributed according to M_0 and M_1. Therefore, the power to discriminate data from P_0 from data from P_1 comes only from the marginals M_0 and M_1.

10) This will be corrected.

11) We solve general f-divergence maximization subject to local privacy conditions (Equation 6) as stated in lines 113-116. The KL divergence maximization problem is a special case presented as a motivating example.

12) This will be corrected.

13) It is true that the definition of staircase mechanisms appears immediately after Theorem 2.1 (in lines 199-205), instead of the beginning of Section 2.1. We will remove the word "first" which is creating the confusion, but would like to emphasize that Theorem 2.1 is self-contained. We will add references to the definition of the staircase mechanisms when it is first used in the proof.

14) Blank

15) The "Contributions" sub-section at the end of the introductory section summarizes the contributions and outlines the remainder of the paper. Given the 10-page limit, a choice has been made not to allocate a section for a conclusion.

16A) Given the space limitations, we had to eliminate a few illustrative figures and examples in the submission. Nevertheless, we still managed to have an illustration (Figure 1 showing the binary mechanism and randomized response) and an example of a staircase pattern matrix (lines 235-239).

16B) The primary contribution of this work is theoretical and the goal is to understand the fundamental tradeoffs between utility and privacy. We agree that adding examples and illustrations to make the paper easier to read is an important part of a longer version of the paper.

Reviewer_3:

1) In the multi-party setting: different parties can collaborate and each party is an analyst. We will clarify this point in the final version.

2) We will comment on this difference in the final version. Note that, using our language, |\mathcalY}| = |\mathcal{X}| for the randomized response.

3) Thanks

4) Theorems 2.3 and 2.6 state that \epsilon^* depends on P_0 and P_1.

5) Thanks

6) The family of staircase mechanisms is universally optimal for all f-divergences, any \epsilon, and any P_0 and P_1. However, the specific optimal staircase mechanism for a given problem might be different, depending on \epsilon, P_0, P_1, and the f.

7) It is an interesting research direction, and one possibility is to use our framework to consider P^*_0,P^*_1 = arg min D_f( P_0,P_1) (where the min is taken over the class of possible distributions).

Reviewer_6:

1) Thanks

2) Correlation changes the problem significantly. For small \epsilon, the desired privacy for X_1 might be impossible since knowing X_2 (for example) already reveals information about X_1. For large \epsilon, one needs to carefully adjust the privacy mechanism according to the joint distribution.

3) We give two explicit mechanism, the binary mechanism and the randomized response, and prove that one can achieve good performance with just these two mechanisms already. In general, one can envision a larger family of mechanisms, which achieve better performance as the family gets more complicated.